# Yes-Associated Protein (Yap) Is Up-Regulated in Heart Failure and Promotes Cardiac Fibroblast Proliferation

**DOI:** 10.3390/ijms22116164

**Published:** 2021-06-07

**Authors:** Maryam Sharifi-Sanjani, Mariah Berman, Dmitry Goncharov, Mohammad Alhamaydeh, Theodore Guy Avolio, Jeffrey Baust, Baojun Chang, Ahasanul Kobir, Mark Ross, Claudette St. Croix, Seyed Mehdi Nouraie, Charles F. McTiernan, Christine S. Moravec, Elena Goncharova, Imad Al Ghouleh

**Affiliations:** 1Heart, Lung, and Blood Vascular Medicine Institute, University of Pittsburgh School of Medicine, Pittsburgh, PA 15213, USA; sanjani@pitt.edu (M.S.-S.); mjb275@pitt.edu (M.B.); tga8@pitt.edu (T.G.A.); jjb100@pitt.edu (J.B.); baojunchang@gmail.com (B.C.); ahasanbd78@gmail.com (A.K.); mctiernanc@upmc.edu (C.F.M.); 2Division of Cardiology, Department of Medicine, University of Pittsburgh School of Medicine, Pittsburgh, PA 15213, USA; 3Davis School of Medicine, University of California, Davis, CA 95616, USA; dgoncharov@ucdavis.edu (D.G.); eagoncharova@ucdavis.edu (E.G.); 4Conemaugh Memorial Medical Center, Internal Medicine Department, Johnstown, PA 15905, USA; malhamay@conemaugh.org; 5Division of Pulmonary, Allergy & Critical Care Medicine, Department of Medicine, University of Pittsburgh School of Medicine, Pittsburgh, PA 15213, USA; nouraies@upmc.edu; 6Center for Biologic Imaging, and Department of Cell Biology, University of Pittsburgh School of Medicine, Pittsburgh, PA 15213, USA; mross@pitt.edu (M.R.); claudette.stcroix@pitt.edu (C.S.C.); 7Kaufman Center for Heart Failure, Department of Cardiovascular and Metabolic Sciences, Cleveland Clinic, Cleveland, OH 44195, USA; moravec@ccf.org; 8Department of Pharmacology and Chemical Biology, University of Pittsburgh School of Medicine, Pittsburgh, PA 15213, USA

**Keywords:** Yap, cardiac fibroblasts, proliferation, heart failure, mTOR

## Abstract

Left ventricular (LV) heart failure (HF) is a significant and increasing cause of death worldwide. HF is characterized by myocardial remodeling and excessive fibrosis. Transcriptional co-activator Yes-associated protein (Yap), the downstream effector of HIPPO signaling pathway, is an essential factor in cardiomyocyte survival; however, its status in human LV HF is not entirely elucidated. Here, we report that Yap is elevated in LV tissue of patients with HF, and is associated with down-regulation of its upstream inhibitor HIPPO component large tumor suppressor 1 (LATS1) activation as well as upregulation of the fibrosis marker connective tissue growth factor (CTGF). Applying the established profibrotic combined stress of TGFβ and hypoxia to human ventricular cardiac fibroblasts *in vitro* increased Yap protein levels, down-regulated LATS1 activation, increased cell proliferation and collagen I production, and decreased ribosomal protein S6 and S6 kinase phosphorylation, a hallmark of mTOR activation, without any significant effect on mTOR and raptor protein expression or phosphorylation of mTOR or 4E-binding protein 1 (4EBP1), a downstream effector of mTOR pathway. As previously reported in various cell types, TGFβ/hypoxia also enhanced cardiac fibroblast Akt and ERK1/2 phosphorylation, which was similar to our observation in LV tissues from HF patients. Further, depletion of Yap reduced TGFβ/hypoxia-induced cardiac fibroblast proliferation and Akt phosphorylation at Ser 473 and Thr308, without any significant effect on TGFβ/hypoxia-induced ERK1/2 activation or reduction in S6 and S6 kinase activities. Taken together, these data demonstrate that Yap is a mediator that promotes human cardiac fibroblast proliferation and suggest its possible contribution to remodeling of the LV, opening the door to further studies to decipher the cell-specific roles of Yap signaling in human HF.

## 1. Introduction

Heart disease and subsequent heart failure (HF) is the major cause of death worldwide [1]. HF primarily depends on left ventricular (LV) remodeling that leads to cardiac dysfunction and stiffness associated with extensive fibrosis [2]. Therefore, elucidation of molecular mechanisms underlying cardiac fibrosis is essential for understanding HF pathobiology and for development of potential therapeutic strategies targeting this condition.

The HIPPO pathway is a growth suppressor signaling pathway that prevents organ overgrowth by inhibiting cell proliferation and inducing differentiation or apoptosis [3]. The key reciprocal downstream effectors of HIPPO are transcriptional co-activators Yes-associated protein (Yap) and Taz, which are positive regulators of cell proliferation and survival [3]. Yap/Taz are negatively regulated by the central HIPPO components, large tumor suppressor 1 (LATS1) and LATS2, via direct phosphorylation, which reduces Yap protein levels by targeting it for degradation [3]. Further, HIPPO pathway is suggested to collaborate with mammalian target of rapamycin (mTOR) in organs and cells and to enhance proliferation, migration and inhibition of cell differentiation [4,5,6,7,8].

Cardiomyocyte-specific gain- and loss-of-function studies demonstrated that Yap promotes embryonic cardiomyocyte proliferation and supports cardiac homeostasis in adult mice by protecting cardiomyocytes from apoptosis and preventing heart fibrosis [9,10,11,12], and as such pharmacological inhibition of HIPPO has been considered an attractive strategy to induce cardiomyocyte proliferation and regaining of heart function after HF. Interestingly, recent reports show Yap as a promoter of cardiac scar formation post cryoinjury using Yap mutant adult mice [13] and a mechanoactivated coordinator that drives a profibrotic response in lung fibroblasts [14]. However, while cardiac fibroblasts are the most abundant cell type in the heart [15] and contribute to cardiac fibrosis via increased proliferation, extracellular matrix production and replacement of cardiomyocytes with fibrotic scar tissue [16,17], the role of Yap in these cells remains to be elucidated.

In the current study, we aimed to determine the expression pattern of Yap in cardiac tissues from patients with LV HF. We further examined the role of Yap in proliferation and pro-fibrotic activation of human adult cardiac fibroblasts. Our data show that Yap appears to predominantly accumulate in cardiac fibroblasts of HF patients and that LV tissues from these patients present with down-regulation of signaling of the Yap major upstream inhibitor LATS1. We further provide evidence that cardiac fibroblast Yap is upregulated under pathophysiological stimuli and identify its role in human cardiac fibroblast proliferation.

## 2. Results

### 2.1. Cardiac Accumulation of Yap in LV of Patients with HF

HIPPO-Yap/Taz signaling plays an important role in cardiac growth of embryos as well as adult mouse cardiac cells and tissue remodeling post injury [9,10,11,12]. However, the status of Yap in human cardiac tissue has not been studied. To determine the status of Yap and Taz in failing heart, we compared LV tissues from non-diseased (control) subjects and patients with LV HF. We found that Yap protein levels were significantly elevated when comparing LV tissues from HF cohorts to controls (Figure 1A,B). These observations in LV tissues were associated with reduced activation of Yap negative regulator LATS1, evidenced by decreased T1079 phosphorylation (P)/total LATS1 (Figure 1C,D). Importantly, immunohistochemical analysis showed that Yap is accumulated predominantly in inter-cardiomyocyte areas (Figure 1E, left panel), where cells positive for fibroblast-specific vimentin (Figure 1E, right panel) are mainly located, strongly suggesting cardiac fibroblasts as a cell type of Yap accumulation in LV of patients with HF.

### 2.2. Combined Exposure to TGF-β and Hypoxia Promotes Yap Accumulation in Human Adult Cardiac Fibroblasts

Analysis of cardiac samples from our HF patients revealed increased protein levels of connective tissue growth factor (CTGF), a regulator of cardiac extracellular matrix and fibrosis [18,19] and a downstream effector of Yap transcriptional activity [20,21,22,23] (Figure 1F,G). To interrogate causality, we moved to mechanistic *in vitro* studies. Failing hearts are subject to regional hypoxia [24], which in turn induces pro-fibrotic cellular changes in fibroblasts [25,26]. Further, the TGF-β1 pathway in the heart contributes to cardiac fibrosis [27] and to exacerbation of the fibrotic response [24,27,28,29,30,31,32]. This supports that both hypoxia and TGF-β1 may act as pro-fibrotic factors during HF. Therefore, we tested whether TGFβ and hypoxia alone or in combination promote Yap accumulation in human primary cardiac fibroblasts. While either stimulus displayed a trend towards increased Yap protein content, the combined TGFβ and hypoxia (TGFβ/hypoxia) treatment significantly elevated Yap protein levels by ~5-fold (Figure 2A,B). Similarly, there was a trend towards decreased LATS1 T1079 phosphorylation with TGFβ treatment but not hypoxia exposure (Figure 2C,D). However, the combined TGFβ/hypoxia resulted in a significant reduction in LATS1 T1079 phosphorylation compared to baseline controls (Figure 2C,D). Based on these data, the combined treatment was selected for further experiments.

### 2.3. Yap Is Required for TGFβ/Hypoxia-Induced Cardiac Fibroblast Proliferation and Akt Phosphorylation, But Not Collagen I Production

As cardiac fibroblasts are the main source of collagen production and extracellular matrix deposition in the heart [33], we first evaluated collagen levels in human cardiac fibroblasts treated with TGFβ/hypoxia. Our analysis demonstrated significantly higher collagen I levels after TGFβ and hypoxia treatment (Figure 3A,B). Given the concomitant increase in Yap and collagen I levels under combined treatment, we investigated the potential role for Yap as an upstream promoter of fibroblast collagen accumulation under the HF-related stimuli TGFβ/hypoxia by depleting Yap in human cardiac fibroblasts using adenoviral delivery of shRNA (Appendix A) [7]. As seen in Figure 3A,B, Yap shRNA had no effect on collagen I levels under TGFβ/hypoxia. Since Yap also acts as a pro-proliferative agent, we next examined effects of Yap on the proliferation of cardiac fibroblasts. Notably, in addition to elevating Yap protein levels (Figure 2A,B), combined TGFβ/hypoxia treatment, but not either treatment alone, also induced a significant increase in cell proliferation, as assessed by cell count assay (Figure 3C). To test for a causal relationship between cell proliferation and Yap, we silenced Yap employing specific siRNA *in vitro* in human primary adult cardiac fibroblasts (knockdown efficiency shown in Appendix A). Yap depletion significantly decreased TGFβ/hypoxia-induced fibroblast cell counts compared to the corresponding control siRNA (Figure 3D). Silencing YAP using an alternative technique with shRNA-infected cells showed similar findings, confirming the role of YAP in proliferation (Appendix A). Combined, these data demonstrate that Yap plays a role in cardiac fibroblast proliferation but not collagen I production.

To further delineate the mechanism of Yap-dependent cardiac fibroblast proliferation, we investigated the role of cell signaling mediator protein kinases Akt in the process as it is implicated in HF and is causally linked to cell proliferation in multiple cell types [25,34,35]. Indeed, our data revealed that Akt phosphorylation at S473 (p-Akt^S473^) and phospho/total Akt ratio (Akt activation) were significantly higher in LV tissues from patients with HF compared to controls (Figure 4A,B and Appendix A), supporting translational relevance for a role for Akt in HF. Interestingly, treatment of human cardiac fibroblasts with TGFβ/hypoxia led to no change in the activation of p-Akt^Ser473^ (Figure 4C,D and Appendix A) and p-Akt^Thr308^ (Figure 4E,F and Appendix A), two main Akt phosphrylation sites essential for its activity [36], at 48 h. This observation was also along with our finding that two earlier time points of 12 and 24 h showed no change in p-Akt^Ser473^ activation (Appendix A–D). In contrast, at 72 h exposure, there was a significant increase in Akt activation at both Ser473 (Figure 4G,H) and Thr308 (Figure 4I,J) and no detectable associated increase in Akt protein levels (Appendix A). Importantly, Yap siRNA reduced Akt activation at Ser473 (Figure 4G,H) and Thr308 (Figure 4I,J) in cells exposed to the TGFβ/hypoxia for 72 h with no significant effect on the Akt protein levels (Appendix A), indicating that Yap acts as an upstream regulator of Akt activation in cardiac fibroblasts under profibrotic, pro-proliferative stimuli.

### 2.4. Combined Exposure of Human Adult Cardiac Fibroblasts to TGF-β and Hypoxia Reduces S6 Activation Independent of Yap

Recent studies report crosstalk between HIPPO signaling and mTOR pathway [7,37], and it is shown that Akt is one of mTORC1 activators in cardiac fibroblasts post ischemia [38]. Further, as one of the important functions of mTORC1 is to regulate protein translation through phosphorylation of downstream effector S6 ribosomal protein [39], which could embody various pathological states, and that mTOR/p70S6 kinase (p70S6K) pathway is involved in cardiac remodeling [40,41], we looked into phosphorylated state of S6 (phosphorylated Ser235/236) and p70S6K as well as levels of mTORC1 complex main subunit, mTOR, and its core component that regulates substrate recruitment and subcellular localization Raptor [42], under hypoxia/TGFβ treatment in human cardiac fibroblasts. Interestingly, our data demonstrated an ameliorated activation of S6, as indicated by Ser235/236 phosphorylation, with no significant effect on total S6 protein level (Figure 5A,B and Appendix A) in response to hypoxia/TGFβ, but also a reduction in phosphorylation of p70S6KT389 (Figure 5C,D). Our observations were that Yap silencing had no effect on these signals, despite a reduction in P-p70S6K T389 at baseline (Figure 5C,D). Moreover, there was no associated change in mTOR and Raptor protein expression levels in any of the groups (Figure 5E–H). A further analysis of mTOR activity via 4EBP1Thr37/46 and mTOR phosphorylation revealed no alteration under hypoxia/TGFβ stress (Figure 5I–L). Together, these data suggest parallel signaling of Yap-Akt axis and mTORC1 in human cardiac fibroblasts under TGFβ and hypoxia co-exposure, and provides preliminary support for a possible involvement of mTORC2 pathway, which requires further studies.

### 2.5. Yap Is Not Required for TGFβ/Hypoxia-Induced Cardiac Fibroblast ERK Activation

As ERK1/2 signaling is additionally implicated in HF and it induces cell proliferation as well as being mechanistically linked to HIPPO/Yap signaling [34,43,44,45], we investigated the role of ERK1/2 in Yap-dependent cardiac fibroblast signaling. ERK1/2 phosphorylation and its activation, but not total protein levels, (Figure 6A,B and Appendix A) were elevated in LV tissue from HF patients compared with non-failing controls. *In vitro* exposure of human cardiac fibroblasts to TGFβ/hypoxia decreased ERK1/2 phosphorylation but had no significant effect on its activation (phosphorylated/total) ratio or protein level (Figure 6C,D and Appendix A) after 48 h. However, TGFβ/hypoxia increased ERK1/2 phosphorylation and activation but not protein level (Figure 6E,F and Appendix A) after 72 h of exposure when compared with vehicle control. This was while no ERK1/2 activation was observed at early time points of 12 and 24 h (Appendix A). Interestingly, in contrast to Akt, ERK1/2 activation was not alleviated by silencing of Yap at the 72 h timepoint and there was even a trend towards increase (Figure 6E,F).

Taken together, these data suggest that Yap is a selective upstream activator of Akt, but not ERK1/2 or mTOR signaling under TGFβ and hypoxia, supporting a role for a Yap-Akt signaling axis in promoting proliferation of adult cardiac fibroblasts while having no apparent effect on collagen I production, mTOR or ERK1/2 signaling.

## 3. Discussion

In this study, we report for the first time that (i) Yap signaling is enhanced in LV from patients with non-ischemic HF, (ii) Yap is induced in human adult cardiac fibroblasts by the pro-proliferative pro-fibrotic TGFβ/hypoxia exposure and (iii) silencing Yap expression inhibits cardiac fibroblast proliferation potentially via down-regulation of Akt signaling but, (iv) without affecting mTORC1 signaling. This work provides a basis for future studies on the role of cardiac fibroblast signaling in the pathogenesis of HF, paving the way for development of new treatment avenues for this progressive disease.

Yap and its upstream negative regulator HIPPO are confirmed modulators of cardiomyocyte growth and survival [46,47] while HIPPO signaling deficiency prevents cardiac fibroblast differentiation during heart development via Yap [48]. Further, Yap is activated in lung fibroblasts, in which it promotes proliferation and expression of profibrotic genes [14]. Importantly, very little is known regarding the role of cardiac fibroblast Yap in the adult heart. In the current study we found increased accumulation of Yap in whole LV tissue homogenates of patients with HF compared to control cohort that was accompanied by a down-regulation of HIPPO central protein kinase, LATS1, the upstream negative regulator of Yap. Further investigation using immunohistochemical analyses showed that Yap is accumulated predominantly in cardiac fibroblasts, suggesting this cell type as a new cellular origin of Yap during HF and an important source for disease-related Yap modulation.

Given the established pro-proliferative role of Yap in other cell types and confirmed link between cardiac fibroblast proliferation, extracellular matrix production and HF pathogenesis [15,16,17], we further investigated potential roles for Yap and events that may initiate Yap up-regulation in adult cardiac fibroblasts. We found that TGFβ and hypoxia, well-recognized drivers of fibrotic cardiac remodeling [25,26,27], promote LATS1-Yap dysregulation in adult cardiac fibroblasts, similar to our observations in HF patients’ cardiac tissues. Up-regulation of Yap was associated with increased cardiac fibroblast growth suggestive of a pathological role for cardiac fibroblasts-specific Yap up-regulation during HF and its potential role in cardiac fibrosis. The ability of TGFβ and hypoxia to elicit cardiac fibroblast-specific Yap expression also highlights the potential pathophysiological role of Yap in modulating fibroblast behavior.

We found that depletion of endogenous Yap protects adult cardiac fibroblasts from TGFβ/hypoxia-induced proliferation. Of note, the pro-proliferative role of Yap had also been reported in pulmonary fibroblasts from fibrotic lungs [14], in agreement with our findings and suggestive of potential similarity of the mechanisms regulating lung and heart fibrosis. Interestingly, Yap appeared to have no significant effect on collagen I production in our study. However, we could not exclude the possibility that increased collagen I accumulation and subsequent associated matrix stiffness may be one of the upstream triggers of Yap induction in cardiac fibroblasts during HF. Indeed, Yap is regulated by extracellular matrix composition and stiffness, and stiff matrix induces Yap and Taz accumulation in pulmonary fibroblasts [14]. Together with our data showing induction of Yap by co-exposure to TGFβ and hypoxia, these findings may suggest a multi-factorial mechanism for Yap regulation in cardiac fibroblasts, which requires further investigation.

Yap is known to promote activation of Akt [37], which, while supporting cardiomyocyte survival [11], is positively linked to increased morbidity and mortality due to cardiovascular complications. Although additional studies are needed to clarify the reported controversies, observed differences may be explained by the cell type-specific role of Yap. Indeed, in addition to reported beneficial pro-proliferative and pro-survival role in cardiomyocytes, in our studies Yap supported both Akt activation up-regulation and pathological growth of cardiac fibroblasts, a major event that contributes to cardiac fibrosis during HF. Notably, induction of S473 phosphorylation, but not protein expression of Akt and concomitant reversal by Yap silencing, occurred after 72 h of TGFβ/hypoxia treatment but not 48 h. While this is demonstrative of a causal relationship between Yap and Akt, it also suggests a delayed response that may be indicative of the involvement of additional yet-to-be-identified late-onset mediators or shift in actions of phosphatases such as PTEN and SHIP downstream of Yap. Interestingly, we found that Yap is not required to sustain ERK1/2 phosphorylation in adult cardiac fibroblasts. These data support specificity of Yap toward Akt, and are in agreement with previous studies on uveal melanoma showing that Yap and ERK1/2 may act as parallel signaling pathways [49].

Further, Yap plays a mediatory role in crosstalk between HIPPO and mTOR [37] and in mTOR activation, which activates AKT signaling, contributing to fibroblast proliferation under Angiotensin II and epidermal growth factor (EGF) stimuli [50,51]. We tested for mTOR and its markers of activity, phosphorylated S6 ribosomal protein (P-S6), P-p70S6K, P-4EBP1, and P-mTOR. Interestingly, while Akt phosphorylation at both Ser473 and Thr308 were increased with TGFβ/hypoxia exposure, we observed a Yap-independent decrease in P-S6 and P-p70S6K with no effect on mTOR and raptor (an mTORC1 core component) protein expression, nor P-4EBP1 and P-mTOR. These data are suggestive of parallel independent mTOR and Yap signaling in adult cardiac fibroblasts, and are consistent with an association between decreased S6 activation and increased cell number, in line with previous studies in embryonic fibroblasts from mice with mutated S6 serine residues which showed accelerated cell division [52]. Alternatively, it is also possible that Yap-mediated Akt activation may occur via mTORC2 signaling in cardiac fibroblasts given that phosphorylation of Akt at S473 is also regulated by mTORC2 [53], an intriguing prospect that requires further studies.

## 4. Materials and Methods

### 4.1. Human Tissue Samples

Deidentified snap frozen LV tissue specimens from non-failing (control) hearts from male Caucasian subjects and LV from gender-matched non-ischemic HF patients whom had undergone heart transplant were provided by Cleveland Clinic Human Heart Tissue Bank and University of Pittsburgh, Pittsburgh Heart, Lung and Blood, Vascular Medicine Institute in compliance with protocols approved by University of Pittsburgh and Cleveland Clinic institutional review boards.

### 4.2. Immunofluorescence

Immunofluorescence analysis was performed as described previously [54,55]. Briefly, cryostat sections (5–7 µm) were washed 3 times with phosphate buffered saline (PBS), then 3 times with 0.5% BSA in PBS, and incubated for 30 min in 2% BSA in PBS. After incubation for 1 h with primary anti-Yap, -vimentin, -Phalloidin, -troponin, and then for 1 h with fluorescence-tagged secondary antibodies (Cell Signaling, Danvers, MA, USA), slides were mounted using Gelvatol mounting media. F-actin was visualized by rhodamine-tagged phalloidin stain (source), cell nuclei with Hoescht dye (bisbenzamide 1 mg/100 mL water). Tissues of three patients with LV HF and three non-failing heart (control) subjects were examined. Images were captured with an Olympus Fluoview 1000 confocal microscope 2.01 (Bethlehem, PA, USA) and ZOE™ Fluorescent Cell Imager (Bio-Rad, Hercules, CA, USA) with appropriate filters.

### 4.3. Immunoblot

Immunoblot analysis was performed as described previously [54,55]. Briefly, LV tissue homogenates and human cardiac fibroblast cell lysates were prepared using ice-cold lysis buffer consisting of 2.3 g HEPES, 1.7 g NaCl, 1.2 g Sodium pyrophosphate, 0.5 g Sodium fluoride, 0.5 g 2-glycerophosphate, 0.2 M Sodium orthovanadate, 0.5 M EDTA and Protease Inhibitor Cocktail (Roche, Basel, Switzerland). Homogenization/lysis was followed by 7 min of centrifugation at 13,500 rpm and the supernatant stored at −80 °C for future analysis. Protein extracts were separated on 10% Bis-Tris gels in parallel followed by transfer of proteins to a nitrocellulose membrane (Invitrogen, Carlsbad, CA, USA). The membranes were blocked for 1 h using 5% dry milk or Licor blocking buffer (Licor, Lincoln, NE, USA) and probed with anti-Yap, -Hsp90, -Erk, and -Phospho (P)-Akt^Ser473^, (P)-Akt^Thr308^, -Lats, -P-Lats, -beta-tubulin, -CTGF, -S6, -P-S6, -P-P70S6K, -P-4E-BP1, -Raptor, -mTOR, -P-mTOR and -GAPDH rabbit, anti -Akt and -P-Erk mouse (1:1000, Cell Signaling, Danvers, MA, US), anti-collagenI goat (SantaCruz, Dallas, TX, USA) and anti-tubulin rat (abcam, Cambridge, MA, USA) overnight at 4 °C. Membranes were then incubated with fluorescent (1:6000, Licor, Lincoln, NE, US) or HRP-linked (1:1000, Cell Signaling) secondary antibodies at room temperature for 1 h. The images were scanned on an Odyssey system imager, developed on film or BioRad ChemiDoc Imaging system and relative band intensities were quantified by densitometry using ImageJ (NIH).

### 4.4. Primary Cell Culture of Human Cardiac Fibroblasts

Human adult ventricular fibroblasts were purchased from LONZA (CC-2904, Walkersville, MD, USA), plated, and expanded according to manufacturer protocol using fibroblast medium supplemented with FGM-3 Bullet Kit LONZA (Walkersville, MD, USA). Cells were exposed to TGFβ (10 ng/mL) and/or hypoxia (1% O_2_) for indicated times. For gene silencing, Yap shRNA-producing or control adenovirus (Origen, Rockville, MD, USA) was transduced into cells [7] for some protocols. For remaining protocols siRNA against Yap or corresponding scrambled siRNA control were transfected using lipofectamine 3000 according to manufacturer’s protocol (Thermofisher Scientific, Waltham, MA, USA). Prior to experiments, cells were serum starved by incubation overnight in basal LONZA medium supplemented with 0.1% BSA. For cell transfections, additional incubation in serum-free medium supplemented with 0.1% BSA was performed during exposure (4–6 h) as recommended in manufacturer’s protocol. Immunoblot analysis was performed as described above and previously published [54,55].

### 4.5. Cell Growth Assay

Cell growth assay was performed as described previously [55]. Briefly, cells were plated on 60 mm cultured plates (78,000 cells/plate), incubated for 4–6 h in serum-free media supplemented with 0.1% BSA and exposed to TGFβ (10 ng/mL) and hypoxia (1% O_2_) separately or in combination. Cell counts were performed using the automated Coulter Counter Z1 (Beckman Coulter, Indianapolis, IN, USA). A minimum of three repetitions per experimental condition were performed in each experiment.

### 4.6. Data Analysis

Statistical analyses were performed using GraphPad Prism (La Jolla, CA, USA). Comparisons between two groups and multiple groups were performed using Mann-Whitney and Kruskal-Wallis with Dunn’s test, respectively. A *p*-value of <0.05 was used as a criterion for statistical significance. Data are presented as means ± SEM.

## 5. Conclusions

Our study provides evidence that Yap is up-regulated in human LV HF and supports pathological cardiac fibroblasts proliferation. Given that pharmacological inhibition of HIPPO and up-regulation of Yap are currently under consideration as an attractive option to induce cardiomyocyte regeneration in HF, our study suggests that pro-proliferative function of Yap in cardiac fibroblasts might be also considered as a potentially unfavorable outcome of HIPPO/Yap-targeted therapy. Further studies are needed to delineate the cell-specific roles of YAP and therefore educate potential future therapeutic strategies.

## Figures and Tables

**Figure 1 ijms-22-06164-f001:**
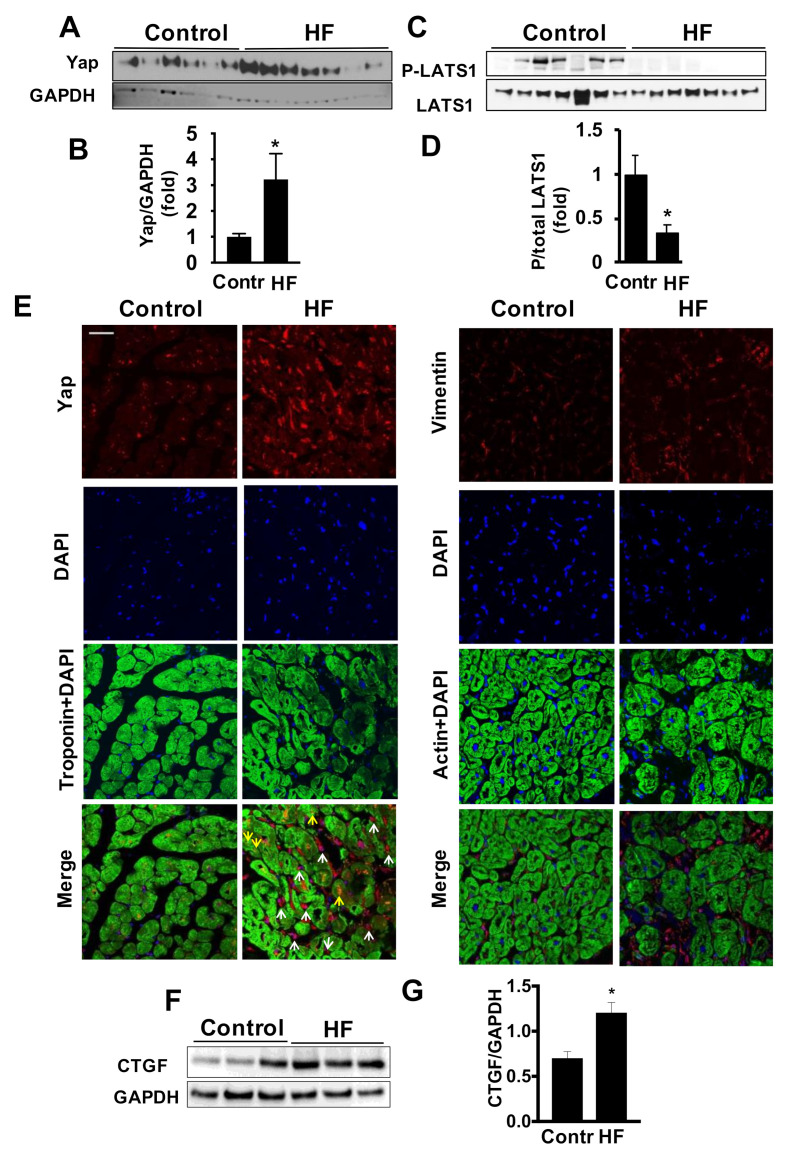
HIPPO signaling is dysregulated in LV of HF patients. (**A**,**C**,**F**) LV tissues from non-failing subjects (control) and patients with HF (non-ischemic, male, Caucasian) were subjected to immunoblot analysis to detect indicated total or phosphorylated (P) protein levels. (**B**,**D**,**G**) Graphs presenting fold change quantitative densitometry analysis of indicated proteins normalized to corresponding loading or total-protein control. Data represents fold changes from control. Data are mean ± S.E.M from 6–9 subjects/group. * *p* < 0.05. (**E**) Immunofluorescent staining analysis of YAP localization (left panel) and vimentin for fibroblast localization (right panel) in human LV cardiac tissues from non-failing (control) and patients with HF. Yap localization to cardiomyocytes and inter-cardiomyocyte areas are identified with yellow and white arrows, respectively. Images are representative from 3 subjects/group. Bar equals 50 µm.

**Figure 2 ijms-22-06164-f002:**
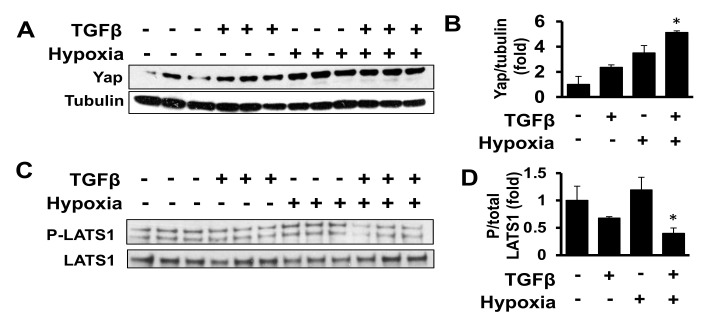
Combination of TGFβ and hypoxia down-regulates LATS1 phosphorylation and promotes Yap protein expression. (**A**–**D**) Immunoblot analysis of indicated proteins for human ventricular cardiac fibroblasts exposed to vehicle control, hypoxia (1% O_2_) and TGFβ (10 ng/mL), separately or in combination, for 48 h (**A**,**B**) and 24 h (**C**,**D**). (**B**,**D**) Graphs presenting fold change quantitative densitometry analysis of indicated proteins normalized to corresponding loading or total protein control. Data represents fold changes from no treatment control (−). Data are mean ± S.E.M; * *p* < 0.05 vs. control (−), *n* = 3/condition.

**Figure 3 ijms-22-06164-f003:**
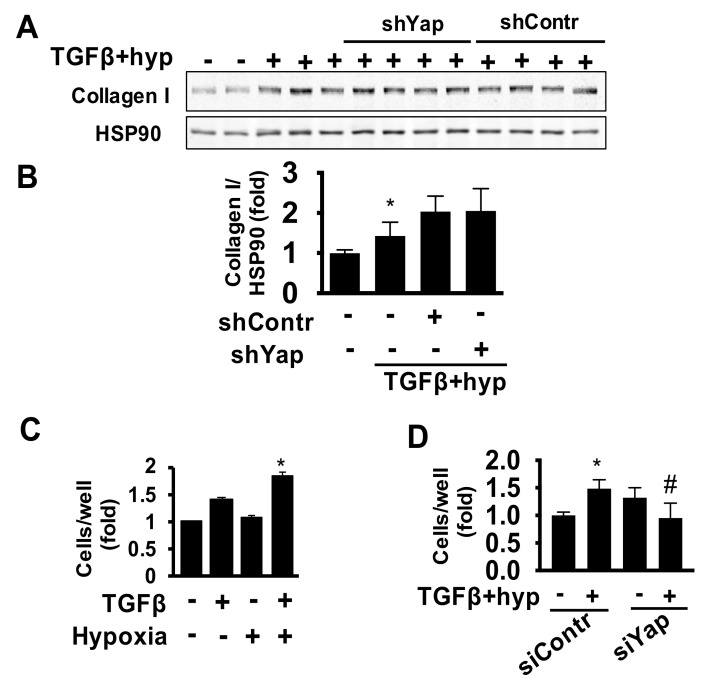
Yap does not contribute to profibrotic stress-related human adult cardiac fibroblast collagen I production but is required for human adult cardiac fibroblast growth. (**A**) Immunoblot analysis to detect collagen I with HSP90 as loading control in cardiac fibroblasts exposed to vehicle (−) or hypoxia (1% O_2_) and TGFβ (10 ng/mL) for 48 h with and without transduction with adenovirus expressing shYap or control shRNA (shContr). (**B**) Graph representing quantitative densitometry analysis of collagen I normalized to corresponding loading control. Data are mean ± S.E.M of fold changes from vehicle-treated cells; * *p* < 0.05 vs. control (−), *n* = 5–8. (**C**) Cell growth assessment by total live cell number of human cardiac fibroblasts exposed to hypoxia (1% O_2_) and TGFβ (10 ng/mL) separately or in combination; cell counts were performed 7 days post-exposure; * *p* < 0.05 vs. control (−); *n* = 3/condition. (**D**) Cardiac fibroblasts transfected with control (siContr) or Yap (siYap) siRNA and exposed to hypoxia (1% O_2_) and TGFβ (10 ng/mL) followed by cell count performed at day 5 of exposure. Data are mean ± S.E.M; * and # *p* < 0.05 vs. siContrl and siContrl+hypoxia/TGFβ, respectively, *n* = 3–4/condition.

**Figure 4 ijms-22-06164-f004:**
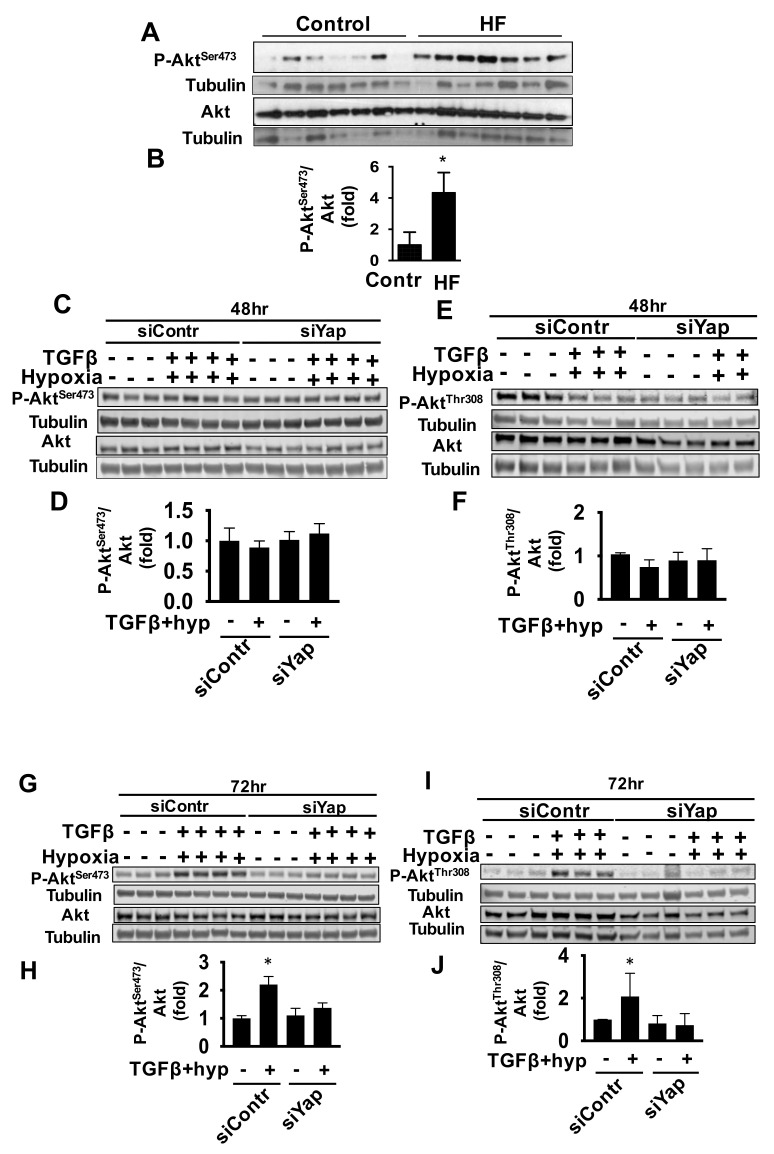
Akt activation is increased in HF and profibrotic conditions in a Yap-dependent mechanism. (**A**,**B**) Representative immunoblot (**A**) and densitometry analysis (**B**) of LV tissues from HF patients and non-failing controls for indicated proteins. P-Akt: phospho-Akt. Quantitative densitometry measured as fold change normalized to corresponding loading control. Data represent mean ± S.E.M of fold changes from control; *n* = 7 subjects/group, * *p* < 0.05 vs. control. (**C**–**J**) Human cardiac fibroblasts transfected with siContr or siYap were exposed to normoxia and vehicle (−) or hypoxia (1% O_2_) and TGFβ (10 ng/mL) for stated durations, and immunoblot analyses were performed to detect indicated proteins. (**D**,**F**,**H**,**J**) Graphs representing fold change quantitative densitometry analysis of total to phosphorylated ratio of the identified protein. Data represent mean ± S.E.M. of fold changes from siContr control (−); * *p* < 0.05 vs. siContr control (−); *n* = 3–4/condition.

**Figure 5 ijms-22-06164-f005:**
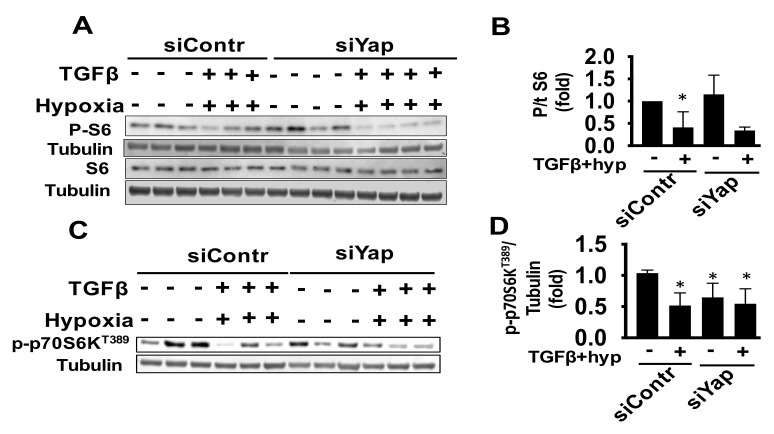
mTOR signaling is decreased under profibrotic conditions through a Yap-independent pathway. (**A**,**C**,**E**,**G**,**I**,**K**) Representative immunoblot for indicated proteins of human adult cardiac fibroblasts transfected with siYap or siContr and exposed to hypoxia/TGFβ. (**B**,**D**,**F**,**H**,**J**,**L**) Fold change immunoblot densitometry analyses of indicated proteins phosphorylated to total ratio or normalized to corresponding loading control. Data represent mean ± S.E.M of fold changes to siContr-infected vehicle-treated control group (−); *n* = 3–4/condition, * *p* < 0.05 vs. corresponding (−) control.

**Figure 6 ijms-22-06164-f006:**
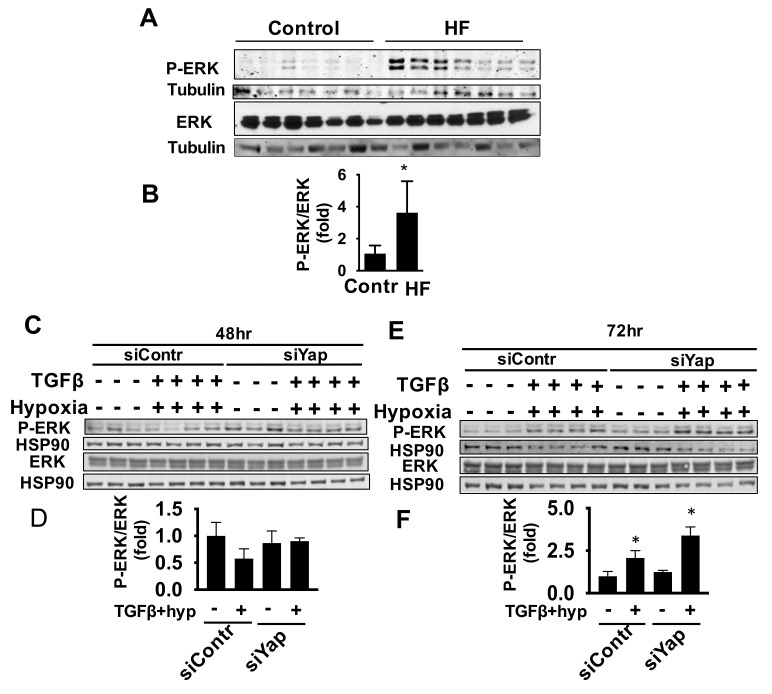
Elevation of ERK1/2 under conditions of HF is not ameliorated upon gene silencing of Yap. (**A**) Representative immunoblot for indicated proteins of LV tissues from HF patients and non-failing controls and (**B**) corresponding densitometry quantitative analyses. P-Erk: phospho-Erk. Data represent mean ± S.E.M of fold changes from control; *n* = 7 subjects/group, * *p* < 0.05 vs. control. (**C**–**F**) Human adult cardiac fibroblasts were transfected with siYap or siContr and exposed to hypoxia/TGFβ for stated durations and immunoblot analyses performed to detect indicated proteins. (**D**,**F**) Graphs representing quantitative densitometry analysis normalized to corresponding loading control. Data represent mean ± S.E.M of fold changes from siContr-infected vehicle-treated control group (−); *n* = 3–4/condition, * *p* < 0.05 vs. corresponding (−) control.

## Data Availability

Data are contained within the herein article.

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
