# Peer review of "Yes-Associated Protein (Yap) Is Up-Regulated in Heart Failure and Promotes Cardiac Fibroblast Proliferation"

_ijms, 2021, doi:10.3390/ijms22116164_

Round 1
Reviewer 1 Report
This paper describes activation of the HIPPO signaling pathway in left ventricular heart failure syndrome resulting in upregulation of the HIPPO pathway effector YAP in tissue of patients with LV HF. Activation is associated with downregulation of LATS and upregulation of the fibrosis marker CTGF. The authors show that YAP upregulation is mainly happening in the cardiac fibroblasts rather than in cardiac myocytes. In addition, treatment of cultured ventricular cardiac fibroblasts with the combined profibrotic stressors TGFβ and hypoxia increased YAP expression, cell proliferation, AKT and ERK1/2 phosphorylation and collagen 1 expression, while decreasing LATS1 activation and ribosomal protein S6 phosphorylation. YAP inhibition under TGFβ/hypoxia stress decreased AKT phosphorylation but had no effect on ERK1/2 or S6 phosphorylation. The authors conclude that YAP is a possible contributor to the remodeling of the LV in LV HF.
Comments:
- Given that the YAP siRNA seems to work much better than the shRNA in figure 3, why didn’t the authors use the siRNA to look at YAP’s effect on collagen production?
- In figure 4 the authors look at AKTSer473 phosphorylation which is increased in fibrotic cardiac myofibroblast and in TGFβ/hypoxia stressed cultured cardiac fibroblasts. This AKT phosphorylation is the result of mTORC2 activity. What happens to AKTThr308 phosphorylation under these conditions, as YAP reduces PTEN expression thereby enhancing the level of PI3K generated PIP3 which stimulates mTORC1 signaling?
- The data presented regarding mTOR signaling in figure 5 are inadequate to draw conclusions. The authors should show p-p70S6KT389 , p4EBP1Thr37/46 and p-mTORS2448 levels to be able to say anything about mTOR kinase activity in the experiments in figure 5. Levels of mTOR protein cannot be used as a measure for mTOR kinase activity. Did the authors try to inhibit mTOR using rapamycin in these experiments and compare that with Raptor knockdown? This is important as Woodcock et al. (Nature Communications 2019) showed that TGFβ induces rapamycin insensitive mTOR signaling via 4EBP1.
- What is the level of gene silencing of YAP in figure 6? The authors should show the level of YAP KD in all figures where they used YAP KD, as was done in figure 3.
Reviewer 2 Report
Sharifi-Sanjani and colleagues reported enhanced levels of the transcriptional co-activator Yap in LV tissue of HF patients and identified Yap as a mediator of cardiac fibroblast proliferation.
Comments:
- Supplemental material is not mentioned in the manuscript.
- Abstract: TGF-β/hypoxia-induced Akt and ERK1/2 activation is not a new finding by these authors. This should be readily apparent in the text.
- Page 3, line 82-90, Fig. 1: The authors conclude that cardiac fibroblast produce Yap, based on their immunostaining showing Yap and a fibroblast marker in the same region of LV tissues. Ideally, a Yap/Vimentin co-staining is needed to draw such a conclusion…
- Figures: Sample numbers are often different. E.g. in Fig. 1 A: control 6, HF: 7, C: control 7, HF: 7, F: control: 3, HF: 3, why?
- Fig. 1E: DAPI staining is hardly visible.
Fig. 1F/G: Differences in band intensity is quite high, but error bars are quite small. What is shown: SD or SEM? This is nowhere indicated in the manuscript.
- Position of figures in the text are sometimes confusing and should be revised. Figures should be placed directly after the text passage where they were described.
- Fig. 2C: Following consequently the chosen experimental setting a housekeeper is missing here (however, in my view this is not absolutely necessary).
- Why are so many different housekeepers used for WB: GAPDH, HSP90, β-Actin, Tubulin…
- Fig. 3D/E: TGF-β/hyp-induced Collagen I is not apparent in this experiment. Experiments should be performed such as in Fig. H.
- Fig. 4,5&6): WB quantification is confusing… the most important quantification which is usually presented is the ratio of the phosphorylated to the not phosphorylated form. Other quantifications could be shown as supplemental graphs.
- Fig. 4&6: Why were these time points investigated (48 and 72 hrs)? It would make more sense to perform time-dependent experiments to identify robust Akt and ERK1/2 activation for subsequent experiments.
